# Case-area targeted interventions (CATI) for reactive dengue control: Modelling effectiveness of vector control and prophylactic drugs in Singapore

Oliver J. Brady[1,2]*, Adam J. Kucharski[1,2], Sebastian Funk[1,2], Yalda Jafari[1,2], Marnix Van Loock[3], Guillermo Herrera-Taracena[4], Joris Menten[5], W. John Edmunds[1,2], Shuzhen Sim[6], Lee-Ching Ng[6], Stéphane Hué[1,2‡], Martin L. Hibberd[7‡]

1 Centre for Mathematical Modelling of Infectious Diseases, London School of Hygiene & Tropical Medicine, London, United Kingdom, 2 Department of Infectious Disease Epidemiology, Faculty of Epidemiology and Public Health, London School of Hygiene & Tropical Medicine, London, United Kingdom, 3 Janssen Global Public Health, Janssen Pharmaceutica NV, Beerse, Belgium, 4 Janssen Global Public Health, Janssen Research & Development, LLC, Horsham, Pennsylvania, United States of America, 5 Quantitative Sciences, Janssen Pharmaceutica NV, Beerse, Belgium, 6 Environmental Health Institute, National Environment Agency, Singapore, Singapore, 7 Department of Infection Biology, Faculty of Infectious Tropical Diseases, London School of Hygiene & Tropical Medicine, London, United Kingdom

‡ Both authors jointly supervised the work.
* oliver.brady@lshtm.ac.uk

**Data Availability Statement:** The data are available from [25], and as part of the DENSpatial R package

## Abstract

### Background

Targeting interventions to areas that have recently experienced cases of disease is one strategy to contain outbreaks of infectious disease. Such case-area targeted interventions (CATI) have become an increasingly popular approach for dengue control but there is little evidence to suggest how precisely targeted or how recent cases need to be, to mount an effective response. The growing interest in the development of prophylactic and therapeutic drugs for dengue has also given new relevance for CATI strategies to interrupt transmission or deliver early treatment.

### Methods/Principal findings

Here we develop a patch-based mathematical model of spatial dengue spread and fit it to spatiotemporal datasets from Singapore. Simulations from this model suggest CATI strategies could be effective, particularly if used in lower density areas. To maximise effectiveness, increasing the size of the radius around an index case should be prioritised even if it results in delays in the intervention being applied. This is partially because large intervention radii ensure individuals receive multiple and regular rounds of drug dosing or vector control, and thus boost overall coverage. Given equivalent efficacy, CATIs using prophylactic drugs are predicted to be more effective than adult mosquito-killing vector control methods and may even offer the possibility of interrupting individual chains of transmission if rapidly deployed. CATI strategies quickly lose their effectiveness if baseline transmission increases or case detection rates fall.

(https://github.com/obrady/SpatialDengue/releases/tag/V1.2).

**Funding:** This work was funded by a grant from Janssen (grant number ITPMZG4810) to MH. OJB [206471/Z/17/Z], AJK [206250/Z/17/Z] and SF [210758/Z/18/ Z] are sponsored by the Wellcome Trust (https://wellcome.ac.uk/). The funders had no role in study design, data collection and analysis, decision to publish, or preparation of the manuscript.

**Competing interests:** I have read the journal's policy and the authors of this manuscript have the following competing interests: MVL, GHT and JM are employees of Johnson & Johnson. LS and LN are employees of the Singapore national Environment Agency

## Conclusions/Significance

These results suggest CATI strategies can play an important role in dengue control but are likely to be most relevant for low transmission areas where high coverage of other non-reactive interventions already exists. Controlled field trials are needed to assess the field efficacy and practical constraints of large operational CATI strategies.

## Author summary

In resource limited settings there is a pressing need for more efficient, more targeted ways of controlling transmission and preventing outbreaks. One option is to use case-area targeted interventions (CATI) that are focussed on areas that have recently reported disease cases. The effectiveness of such CATI strategies is highly dependent on how the disease spreads. Despite CATI strategies being widely used to control the vector-transmitted disease dengue, little evidence underpins its effectiveness.

In this analysis we formulate a mathematical model designed to test the effectiveness of CATI strategies for dengue control in Singapore- a best case test scenario for the approach. Simulation from this model suggested CATI are likely to be effective for dengue, but need to have large (250m+) radii around index cases and may not be suitable in higher transmission areas.

These results, when combined with limited field evidence of efficacy, suggest that CATI strategies are unlikely to be universally applicable dengue control tools. Only once high coverage with other (non-reactive) interventions has been achieved and comprehensive rapid disease surveillance has been established are CATI strategies likely to become efficient methods of disease control.

## Introduction

Dengue is a viral disease vectored by the *Aedes aegypti* mosquito that causes an estimated 100–400 million infections per year worldwide [1, 2]. While most dengue virus (DENV) infections only result in mild disease, the ubiquity of infection causes significant morbidity and frequently overwhelms healthcare facilities. The arboviruses of chikungunya and Zika which are also transmitted by the *Aedes aegypti* mosquito threaten to establish permanent transmission cycles if current dengue outbreaks cannot be rapidly and effectively controlled.

No country has been able to permanently eliminate DENV transmission and focus has shifted towards control and, in particular, preventing big outbreaks as early as possible [3]. One strategy recommended to achieve this is the use of case-area targeted interventions (CATI), where interventions are targeted in and around (typically within 100m) the houses of dengue patients after they present to healthcare facilities [3]. For dengue, these interventions are currently restricted to vector control activities including fogging to kill adult mosquitoes and source reduction to eliminate mosquito egg-laying habitat [4]. CATI strategies are becoming increasingly popular given the need to optimise the use of limited vector control resources and the challenges of sustaining area-wide control [5, 6]. However, the evidence-base for the speed, scale and efficacy of response required for CATI strategies to contain a dengue outbreak remains far from complete. The suggestion that the control radius should equal 100m is largely based on maximum *Ae. aegypti* mosquito flight range estimates from mark-release-recapture studies, which may not cover sufficiently large areas to properly document maximum dispersal [6]. Such studies may also not be representative, and longer flight ranges (>300m) have been documented [7]. In addition, there is strong evidence that urban spread of dengue is largely

attributable to movement of infected humans [5, 8, 9] and response time recommendations are based more on practical realities than quantitative estimates of likely effectiveness [3].

While no drug is currently used for dengue treatment or prevention, a range of short-acting anti-viral compounds are in development that could protect against infection, reduce human infectiousness and decrease the severity of disease [10]. Such prophylactic drugs could play a key role in CATI strategies if their deployment can be optimised and appropriately combined with vector control.

As dengue surveillance systems have become increasingly electronic and automated, there is renewed interest in using real-time CATI strategies for dengue control. However, three features of dengue epidemiology may limit the effectiveness. First, a high proportion of DENV infections are not detected by routine surveillance systems [11], meaning foci of infection may be missed. Second, index case cluster studies have shown that humans peak in infectiousness at the time of, or just before, becoming symptomatic [12] meaning the response may be too slow to prevent onwards transmission. Third, *Ae. aegypti* mosquitoes usually bite during the day when people are more mobile, meaning the place of residence may not accurately predict the location where they were infected.

No randomized control trials have been undertaken for CATI against adult *Ae. aegypti* [6], but there is evidence to suggest that dengue infection is highly spatiotemporally clustered either in the household or within the local area (up to 1km) [5, 13–15]. In the absence of locally applicable field trials, mathematical models can be used to predict what effect response speed and scale will have on the effectiveness of a CATI strategy and identify what role novel prophylactic drugs could play in improving outbreak control. Two theoretical modelling studies of spatially reactively targeted interventions (CATI) for outbreak control have been undertaken for dengue [16, 17], but have only considered simplistic descriptions of human movements between two patches, i.e. travel between work or home or between village and city. One more detailed modelling study for chikungunya [18] suggests that the effectiveness of CATI is likely to be highly sensitive to the proportion of symptomatic cases, which is known to be lower for dengue [1, 19, 20]. Finally, one arbovirus transmission model has been recently used to compare index case-targeted vector control with realistic human movements. This found that the performance of CATI strategies is highly dependent on the human density of the area concerned, but this analysis was limited to more spatially coarse neighbourhood-level analyses [21]. None of these analyses have been formally validated in their ability to reproduce the unique spatial clustering patterns of urban dengue outbreaks and no modelling analyses have considered the potential use of prophylactic drugs for targeted outbreak control.

Here we test the potential effectiveness of vector control and prophylactic drug-based CATI strategies for the control of dengue outbreaks and identify the operational speed and scale of response required. For this, we use a spatially explicit dengue dataset describing a series of outbreaks in Singapore between 2013–2016. Singapore has a robust vector control programme that maintains low baseline transmission intensity, a strong case surveillance system that detects a relatively high proportion of DENV infections in a timely manner and has readily available geopositioned dengue case data over multiple years to characterise transmission [22–24]. These characteristics make such a location a best-case scenario for CATI strategies for dengue control.

## Methods

### Data

We used weekly data on the location of cases across Singapore for the time period May 2013 – June 2016 as reported by the Singapore National Environment Agency (https://www.nea.gov. sg/dengue-zika/dengue/dengue-clusters, S1 Fig). A total of 82,185 geopositioned suspected

and confirmed cases occurred over this time period with peaks in early-to-mid 2013, 2014 and 2016. The data are available from [25], and as part of the DENSpatial R package (https://github.com/obrady/SpatialDengue/releases/tag/V1.2).

To represent seasonal fluctuations in transmission intensity from the mosquito population, data on adult mosquito abundance was obtained from the National Environment Agency entomological surveillance programme over the same time period and is included as part of the DENSpatial R package. The average weekly Gravitrap *Aedes aegypti* index (GAI–number of female aegypti caught per functional trap) across 34 representative sentinel sites measures seasonal abundance across Singapore (SI1.1) [26].

To generate high resolution population maps building-level data from OpenStreet (www.openstreetmap.org) was combined with planning area-level population estimates from the Singapore department of Statistics 2020 population trends report [27]. Areas where humans do not spend significant portions of time were excluded then information of build size and height were extracted to use as a basis for apportioning planning-area level population counts (S1 Text). This population raster was then aggregated to 1km x 1km pixel level then rounded to the nearest integer for all subsequent model-based analyses. This 1km x 1km patch size choice was based on previous work that suggests dengue transmission in large outbreaks tends to be highly spatially clustered up to 1km) [5, 13–15]. For clarity, this does while the model is implemented on a 1km x 1km patch scale, the model is still able to track finer-scale transmission events and intervention policies by assuming individuals homogeneous mix and that features of the environment (mosquito population size and human immunity) are broadly similar at sub 1km x 1km scales. We believe to be an acceptable approximation for assessing the effects of CATI interventions during large outbreaks.

## Model structure

The model used for this analysis is a spatially explicit patch based stochastic model. Within each patch, humans are modelled in Susceptible, Infectious and Recovered compartments (SIR) with infectiousness varying over time since becoming infected (Fig 1). We assume humans peak in infectiousness at the timing of onset of symptoms (i.e., after completion of the intrinsic incubation period (IIP)). Following the IIP they develop either apparent illness and are detectable or remain inapparent until 8 days post symptom onset upon which they transition to the Recovered and immune compartment. Humans can also move to a temporarily immune compartment (Rp) if treated with prophylactic drugs. Mosquitoes also have an additional exposed but not infectious compartment but with no recovery from infection (SEI). To capture realistic time delays in transmission and detection, intrinsic (human) and extrinsic (mosquito) incubation periods are included and infected humans can only be detected as symptomatic cases after completing their IIP. We assume that all humans contribute to transmission the same regardless of their symptoms or whether they were detected as cases.

Seasonal variation in transmission intensity is represented in our model using data from weekly adult mosquito collections (GAI, see data section). Temporal variations in GAI over the relevant time period were scaled to have a mean equal to one and a floor of 0.1 (assuming transmission is possible even at the most unsuitable times of year) then was used to multiply the transmission coefficient for the corresponding day of the year. We assume that outbreaks occur in this setting due to a combination of seasonal effects and more permanent (up to 1 year) changes to local environments that increase human-mosquito contact rates, for example, construction site development [22].

While transmission dynamics are modelled at the patch level, human populations are modelled as continuously mobile, allocating their time to a number of patches including their home

## Transient human population

**Fig 1. Mathematical model structure for each patch.** Humans (h) and mosquitoes (m) are divided into susceptible (S), exposed (Mosquito-only, E), Infectious (I) and recovered and immune (R) compartments. Humans can become temporarily immune ($R_p$) if treated with prophylactic drugs with effective coverage rate $r_d$ with protection waning at rate $c_d$. Infection is controlled by a mosquito-human contact rate ($\beta$) after which humans and mosquitoes go through an incubation period (at rates $\varepsilon_h$ and $\varepsilon_m$ respectively). Humans then naturally recover after $1/\gamma$ days of illness while mosquitoes stay infectious for life. Mosquitoes die at a natural death rate $\mu_n$ but can also be subject to an additional focal control mortality rate $\mu_c$. All human population compartments are transient and made up of all individuals who spend time in the patch. Mosquitoes do not move between patches.

residence. Here we represent flow of humans between patches using three different movement models: i) exponential, ii) gravity and iii) radiation to represent simple distance-based, source-sink and structured commuting-type human movement behaviours respectively [28, 29] (equations in S2 Text). The choice of human movement model is estimated during the model fitting procedure. We assume mosquitoes do not move between patches (patch size 1km x 1km) consistent with the limited dispersal of the primary species vector in Singapore (*Ae. aegypti*) [22].

Despite the co-circulation of multiple serotypes in Singapore both past and present, the outbreaks between 2013 and 2015 were predominantly due to DENV1 (70–90% of reported cases), as were the majority of cases leading up to the period of interest with the exception of a brief period of DENV2 circulation 2007–2012 [30, 31]. With a focus on only modelling the dynamics of the outbreaks in question we assume only a single serotype model with sterilising infection. Given the existence of temporary cross serotype immunity following recent infection with any one dengue type and the predominance of one type, we would expect similar results from a multi-serotype model over the short timeframe over which our model is focussed.

## Transmission dynamics

Mosquitoes within each patch ($i$) exist in one of three states: Susceptible ($S^m$), Infected (but not yet infectious) ($E^m$) and Infectious ($I^m$). Transitions between these states are determined by the following equations:

$$dS^m/dt = (\mu_n + \mu_c)(S^m + E^m + I^m) - \omega S^m - \mu_n S^m - \mu_c S^m$$

$$dE^m/dt = \omega S^m - \mu_n E^m - \mu_c E^m - \varepsilon_m E^m$$

$$dI^m/dt = \varepsilon_m E^m - \mu_n I^m - \mu_c I^m,$$

Where $\mu_n$ and $\mu_c$ are the rates mosquito mortality due to natural and extra CATI measures respectively (i.e., we assume vector control measures reduce the number of adult mosquitoes), $\varepsilon_m$ is the rate of EIP completion and $\omega$ is the risk of infection for each susceptible mosquito in the patch. Calculating $\omega_i$ involves summing the total person hours spent in patch $i$ by both human residents ($T_{i \to i}$) and visitors ($T_{j \to i}$). Infectivity of any given human is a function of the mosquito-human contact rate of patch $i$ on day $z$ ($\beta_{i,z}$) and the individual infectiousness of each human which is dependent on the number of days since becoming infected ($\theta_d$).

$$\omega_i = \sum_j \left[ T_{j \to i} \left( \frac{\beta_{i,z} \theta_d I_{j,d}^h}{N_j^h} \right) \right]$$

Consistent with observations of viral titre in dengue patients [12] over time, we assume normally distributed infectiousness, peaking at symptom onset (where transmission probability = 1), standard deviation of 2 and constrained to 0 at time of infection and eight days post symptom onset. The timing of peak viremia is determined by a lognormal IIP ($\varepsilon_h$) [32].

$$\theta_d = N(\varepsilon_h, 2)$$

$$\varepsilon_h = Lognormal(\mu, \sigma^2)$$

The additional risk of mortality due to CATI mosquito control efforts ($\mu_c$) is applied if the patch ($i$) is within the defined radius ($L$) of an index case ($j$).

$$l_{min} = \min(l_{i \to 1}, \dots, l_{i \to j})$$

$$f(\mu_c) = \begin{cases} \mu_c & \text{if } l_{min} \leq L \\ 0 & \text{if } l_{min} > L \end{cases}$$

Infection dynamics in humans were modelled in Susceptible ($S^h$), Infectious ($I^h$), recovered due to natural infection ($R^h$) and temporarily immune due to prophylactic drugs ($R_p^h$). Transitions between these states are as follows:

$$dS^h/dt = rR_p^h - \varphi_i S^h - c_d S^h$$

$$\frac{dI^h}{dt} = \varphi_i S^h - c_d I^h - I_{d = \frac{1}{\varepsilon_h} + \frac{1}{\gamma}}^h$$

$$dR^h/dt = I_{d = \frac{1}{\varepsilon_h} + \frac{1}{\gamma}}^h + c_d I^h$$

$$dR_p^h/dt = c_a S^h - r R_p^h$$

$$S^h + I^h + R^h + R_p^h = N^h$$

Susceptible humans can either be infected in their resident ($i$) patch or any of the patches they visit ($j$):

$$\varphi_i = \sum_j \left[ T_{i \to j} \frac{\beta_{j,z} I_j^m}{\sum_k (T_{k \to j} N_k^h)} \right]$$

All individuals ($i$) who reside within a distance $L$ of the home location of any detected dengue case ($j$) will receive prophylactic drugs with an effective coverage level $c_d$.

$$l_{min} = \min(l_{i \to 1}, \dots, l_{i \to j})$$

$$f(c_d) = \begin{cases} c_d \text{ if } l_{min} \le L \\ 0 \text{ if } l_{min} > L \end{cases}$$

Effective coverage includes barriers to access, adherence, eligibility and efficacy of the prophylactic drug in question. Individuals who are treated with drugs when already infected are assumed to acquire sterilizing immunity akin to natural infection. Infected individuals remain in the Infectious state for the duration of their IIP + 8 days of disease ($1/\gamma$) before transitioning to recovered state with sterilizing immunity. Individuals can be detected ($D_t^h$) at any point in their symptomatic infectious stage (i.e., $1/\varepsilon_h < d < (1/\varepsilon_h + 1/\gamma)]$) with daily detection probability $\delta$ which is inferred from the case data (N.B. overall probability that a case is detected is therefore $8\delta$):

$$D_t^h = \delta I_{\frac{1}{\varepsilon_h} < d < \left[\frac{1}{\varepsilon_h} + 8\right]}^h$$

$$\varepsilon_{h,d} = Lognormal(\mu, \sigma^2)$$

The effective reproductive number for each patch was equal to the product of the total number of bites an infectious person receives over their duration of their infectious period ($\theta$), the probability each infected mosquito survives beyond the virus' EIP [33] and the number of infectious bites delivered to susceptible humans by the infected mosquito population post EIP:

$$R_{eff} = \frac{\hat{\beta} S^m \theta}{\overline{N^h}} \cdot \frac{\varepsilon_m}{\mu_n + \varepsilon_m} \cdot \frac{\hat{\beta} \overline{S_h}}{\mu_n \overline{N_h}}$$

$$\overline{N^h} = \sum_j (T_{j \to i} N_j^h)$$

$$\overline{S^h} = \sum_j (T_{j \to i} S_j^h)$$

The model is implemented using a daily time step with events modelled as realisations from stochastic binomial processes. Where multiple additions or subtractions from state

compartments existed (e.g., Exposed mosquitoes completing EIP or dying), the order of processes were randomised then sequentially carried out to ensure each mosquito or human had only one outcome and that certain processes were not more likely than others.

## Parameters and priors

Parameter estimates and priors are given in Table 1. We use fixed distributions for parameters describing DENV IIP and EIP ($\varepsilon_h$ and $\varepsilon_m$) from a systematic review of transmission experiments [32]. Our estimate of the maximum duration of symptomatic illness ($1/\gamma$) corresponds to the upper limit of estimates of the length of the febrile phase of illness [3]. For other parameters we assign weekly informative priors that are either uniform ($k, \delta, \beta$) or broadly distributed ($\mu_n$) [34]. The vector-human transmission rate ($\beta$) was split into two components; a mean ($\beta_\mu$) and a degree of spatial correlation with the existing levels of immunity in the population ($\beta_p$). Because our fitting metrics focussed on the relative distribution of cases over space and time, the daily probability of detection of an infectious human ($\delta$) was used as a scaling parameter to match reported case count post-hoc after other parameters had been fitted. No constraints were put on the daily probability of detection other than $0 \leq \delta \leq 1$.

In the absence of an internationally recognised target product profile for dengue prophylactics, here we consider a drug with 90% protective efficacy against infection administered with 90% coverage to all individuals regardless of infection status ($c_d = 0.81$). We assume that the drug also acts as a therapeutic, preventing transmission and symptoms if given to someone already infected with DENV. We assume the drug is taken with sufficient frequency to ensure

**Table 1. Model parameter values.**

| Parameter | Definition | Value | Source |
|---|---|---|---|
| *Fixed parameters and constraints* | | | |
| $1/\varepsilon_h$ | Intrinsic incubation period of the virus | LogNormal($\mu = 5.9, \sigma = 1.05$) | [32] |
| $1/\varepsilon_m$ | Extrinsic incubation period of the virus | LogNormal($\mu = 7, \sigma = 1.23$) | [32] |
| $\theta$ | Cumulative human infectiousness over the course of an infection | $\theta = \int \frac{negBin(29.37,0.88)}{max(negBin(29.37,0.88))}$ | [35], S2 Fig |
| $1/\gamma$ | Maximum duration of symptomatic illness following completion of IIP | 8 days | [3] |
| $R_{t,0}$ | Average effective reproduction number at the beginning of the outbreak | $1 < R_{eff,t=0} < 10$ | [36] |
| $1/r_d$ | Duration of effectiveness of prophylactic drugs and the number of days for which vector control is effective | Assumed 30 | - |
| *Fitted parameters with priors* | | | |
| $k$ | Proportion of time at risk of dengue infection** each human spends at home in patch *i* vs all other locations | $\mathcal{U}(0,1)$ | - |
| $\delta$ | Daily probability of a DENV infected individual being detected and reported as a case | $0 \leq \delta \leq 1$ | - |
| $\mu_n$ | Daily mortality rate of a mosquito | Beta($\alpha = 11.93, \beta = 107.4$) | [34] |
| $\beta$ | Vector-to-human and human-to-vector transmission rate* | $\mu = \mathcal{U}(0.1, 2)$ $p = \mathcal{U}(-1, 1)$ | - |
| $\tau$ | Type of human movement model | "gravity", "radiation" or "exponential" (S2 Text) | - |
| *Experimentally varied parameters* | | | |
| $c_d$ | The effective coverage of prophylactic drugs in the target area | 0.81 (0.2–0.81) | - |
| $\mu_c$ | The effective coverage of adult mosquito vector control in the target area | 0.81 (0.2–0.81) | - |
| $L$ | Radius around an index case that receives drug dosing or vector control | 50–1000 meters | - |

\* Vector-human transmission rate and its variation over space is parameterised using mean ($\mu$) and correlation with baseline immunity ($p$) parameters.

\*\* The total time at risk of dengue infection can equal less than 24 hours e.g., only at risk at dawn and dusk, this parameter merely divides however many hours are at risk among different patches.

30 days of effective prophylaxis with complete adherence. For comparability, we assume that each application of vector control is effective for 30 days, has a 90% probability of killing adult mosquitoes on each day and is applied with 90% coverage across the target area. For index case radii < 1000m we assume a proportional reduction in coverage that lowers effective coverage.

## Initial conditions

A nationally-representative age-stratified serosurvey for dengue was conducted between December 2013 and February 2014 just before the two main focus time periods of our data clusters [37]. In this study Tan et al. found a nationwide average IgG prevalence of 48.6%, with clear increases with age. Despite the authors finding no geographic variation in age-specific IgG prevalence rates, because the average age of residents varies across Singapore, the observed IgG prevalence in the resident population will still vary. To represent this geographic variation we combined the age-specific IgG seroprevalence measurements from Tan et al. with age-stratified population data from Singapore Department of Statistics 2020 population trends report [27] (S3 Fig). Dengue virus serotype 1 has predominated (accounting for 70–90% clinical cases) since Singapore began DENV serotype surveillance with the exception of the years of 2007–2012 [24, 31]. We therefore consider an average IgG seroprevalence of 48.6% representative of the degree of functional immunity to dengue in Singapore in 2014 despite the co-circulation of other serotypes at lower prevalence during this time.

The susceptible mosquito population size was initialized with an inexhaustible 1:1 ratio with human population size. Model fitting was insensitive to initial mosquito population size as the magnitude and spatial variation in mosquito-human contact rate can compensate for any true differences in population size. The number of infected humans in the previous transmission generation was calculated from the first timepoint of the fitting dataset with the total number of infectious and exposed mosquitoes derived using the following equations:

$$I_{0-g}^h = D_0^h / 8\delta R_t$$

$$E_0^m = I_{0-g}^h \theta \beta_z$$

$$I_0^m = E_0^m (1 - \mu_n)^{\left(\frac{1}{\varepsilon_m}\right)}$$

Infectious and exposed mosquitoes were randomly distributed in space weighted by the product of each infectious persons' time allocation between patches and variation in infectiousness given days since becoming infected. Because cases were only reported weekly, we randomly assigned a day to each reported case over the week prior to their notification.

## Model fitting

Because the time series data we had access to began mid-epidemic and because Singapore often sees frequent importation and co-circulation of multiple chains of transmission [38], we pre-processed the data using a data clustering algorithm prior to model fitting. This clustering algorithm grouped reported cases into transmission clusters based on their timing and location. For this analysis a simple two week generation time was assumed [39] and data from three weeks ago was used when data from two weeks prior was missing. Parent cases were identified by finding the nearest case from two weeks ago with distance measured using i) exponential, ii) gravity or iii) radiation model networks (generating three datasets). Cases were considered to belong to a new case cluster (following importation) if the distance of the nearest

case was above the 99[th] centile of case distance measurements. A sensitivity analysis to this choice of linkage threshold was performed at 95[th] and 99.9[th] centiles (S1 Table). The three most probable clusters were selected for model fitting (3 clusters x 3 movement models = 9 fitting datasets). To focus on the dominant period of transmission, each cluster was restricted to either the 2014 (Weeks 40–99) or 2015 (Weeks 100–160) dengue season based on height of the peak.

The model was fitted to each movement model cluster ($x_\tau$) combination using sequential Monte Carlo approximate Bayesian computation (SMC ABC) following an algorithm from Toni et al. [40]. We fit the model by minimising mean squared error of the relative distribution of cases across eight equally spaced time periods throughout the epidemic ($T$) and eight quadrants of Singapore ($Q$):

$$d_{x_\tau} = T^{rel} + S^{rel}$$

$$T^{rel} = \sum_{T=1}^{T=8} (D_T^{h,observed} - D_T^{h,predicted})^2$$

$$S^{rel} = \sum_{Q=1}^{Q=8} (D_Q^{h,observed} - D_Q^{h,predicted})^2$$

Three rounds of SMC were performed with each round consisting of 1,000 parameter samples, each evaluated from the median prediction from 10 model simulations. The top 10% parameter sample combinations (lowest $d$ values) were retained and used for the proposal distribution in the next SMC round. Only combinations of samples that gave an effective reproductive number between 1 and 10 were accepted to increase computation efficiency. For round one, parameter samples were taken from the priors in Table 1. For rounds two and three parameter combinations from the previous round were sampled then individually uniformly perturbed up to ±10%.

## Results

### There is a limited time window for preventing individual transmission events, but a prophylactic drug could extend this

To determine the effective time window for CATI strategies we first constructed a simple model of DENV generation time in one human host and their vectors (Fig 2). This simple model assumed that vector control and prophylactic drugs had 100% efficacy and that delays in their application was the only factor that limited effectiveness. A limiting factor for reactive dengue control is that humans only become symptomatic at the time of, or just before, their peak in viraemia. As dengue is typically a mild illness, especially in the early stages, symptomatic individuals often take several days to seek treatment, receive a correct diagnosis and be notified to the public health authorities (detection delays). Further delays in notification of the case, its details and the organisation of a house/area visit (response delays) can also mean reactive control can be implemented days or even weeks after the transmission event.

We find that even short delays in detection or response can have a significant impact on the probability of controlling onward transmission. The probability of preventing onward transmission by killing adult mosquitoes (with, for example, fogging) rapidly declines with time; a response nine days after symptom onset is approximately half as effective as a response on the day of detection (Fig 2B). By contrast, adult vector control responses deployed greater than

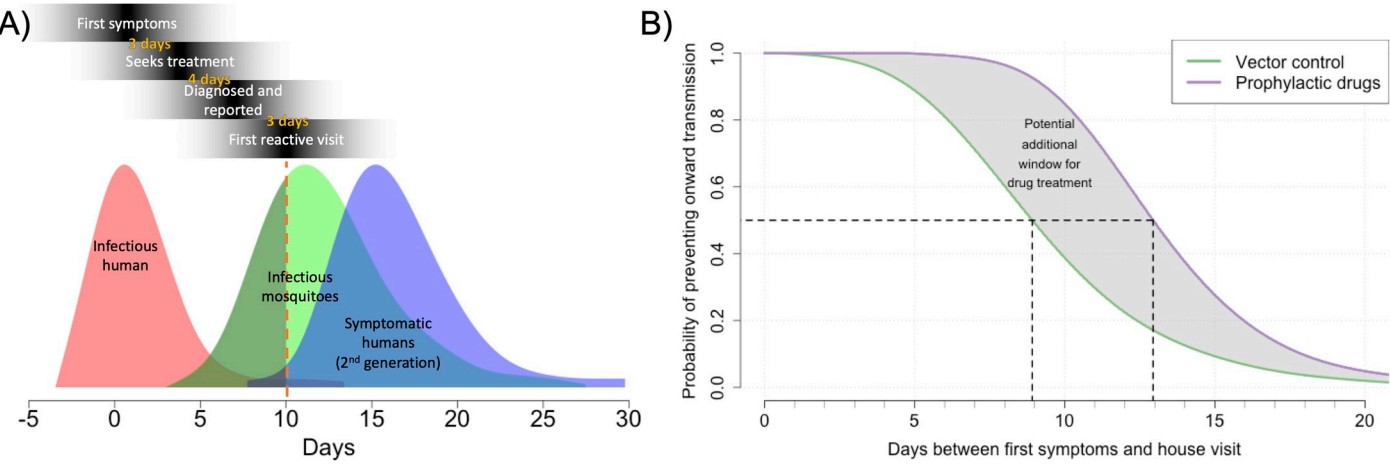

**Fig 2. Probability of interrupting a single chain of transmission over time.** A) the probability distribution of infectious humans, the mosquitoes they infect and the subsequent human dengue cases. The red dotted line shows a typical reactive delay. Day 0 is the day of first symptoms due to dengue. B) Probability of interrupting transmission over different delay durations. Black dotted lines show the time when there is a 50% chance of interrupting transmission.

two weeks after the patient's first symptoms have less than a 20% chance of interrupting onwards transmission.

CATI strategies with prophylactic drugs have the potential to extend this tight time window of effective control by as much as the human IIP (~ 6 days), depending on how such a prophylactic drug prevented disease and transmission. Averting disease cases with reactive control using prophylactic drugs could be effective (> 50% chance of interrupting transmission chains) up to 13 days post symptom onset (Fig 2).

While this simple model gives insight into how reactive delays affect the relative effectiveness of controlling a single chain of transmission, additional factors complicate predicting absolute effectiveness of such strategies. This is because: i) it assumes that control can be accurately targeted to the area(s) of infected mosquitoes (vector control) or infected humans (drugs) which may not be the case as infection events typically happen in the day when many humans may not be at their place of residence and ii) it assumes that all infections can be detected when in reality many dengue infected individuals are asymptomatic or don't seek treatment. The degree of spatial clustering of successive generations of DENV transmission affect the importance of each of these factors. Therefore, to predict effectiveness in more detail we constructed a spatially explicit stochastic model fit to dengue outbreak data from Singapore and simulated the effectiveness of different vector control and drug CATI strategies.

## Model fit results

As a data pre-processing step, we first disaggregated reported cases in Singapore from May 2013 –June 2016 into distinct clusters using a spatiotemporal case-clustering algorithm based on three different kinds of human movement models (Fig 3). There was a consensus among movement models on the emergence of three main case clusters during the period of observation. The largest (primary cluster, red Fig 3) has seen continual low-level circulation until causing a large outbreak in 2015 (peaking at week 140). During the 2014 outbreak two main clusters predominated that then fell to low levels in 2015 (Secondary and Tertiary clusters Fig 3). All cluster algorithms also identified other clusters throughout the period of observation that may represent pre-existing case clusters with an unobserved origin, importations from neighbouring countries or smaller foci of transmission. To simplify model fitting, we chose to

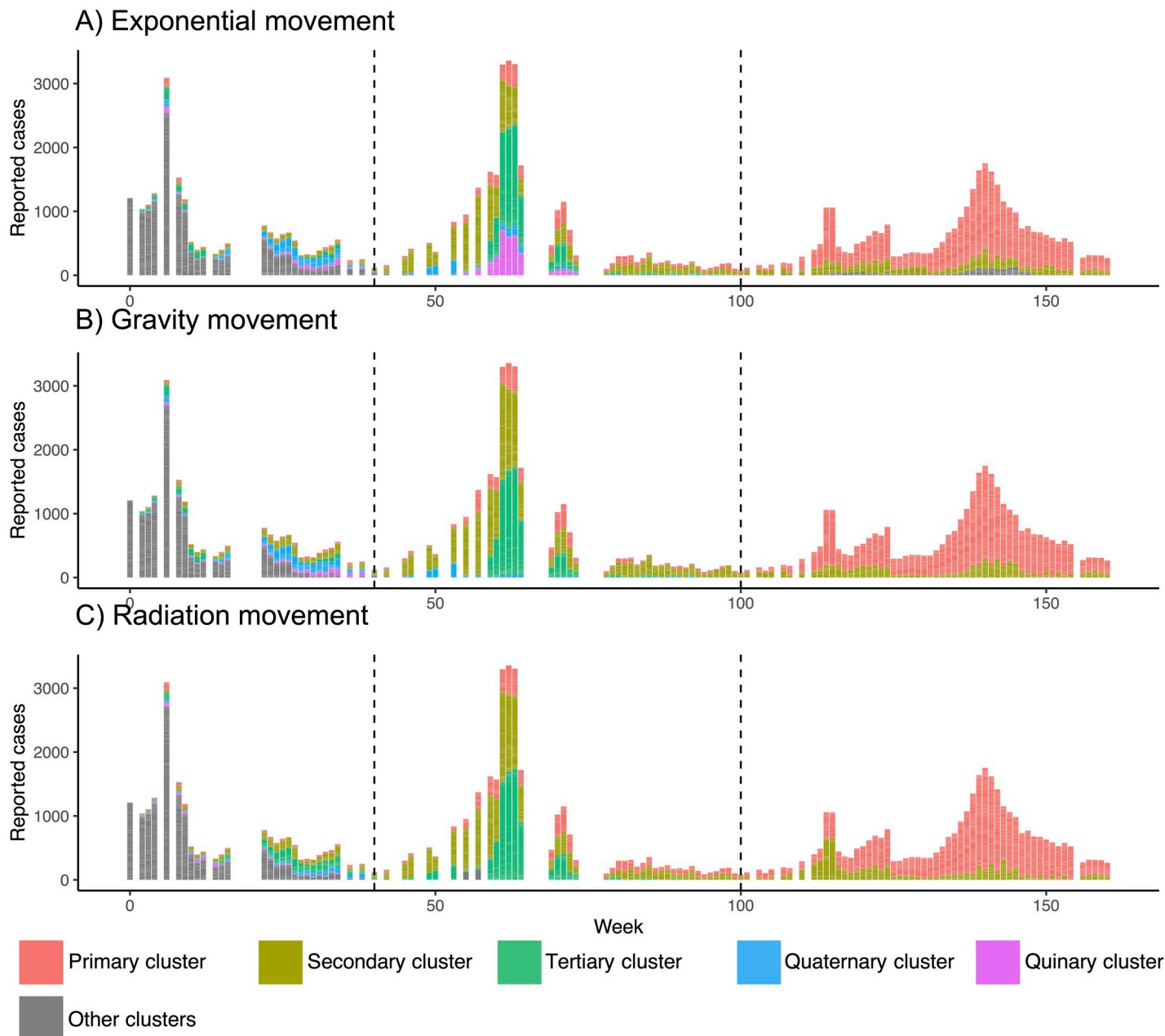

**Fig 3. Top three identified case clusters according to exponential (A), gravity (B) and radiation (C) human movement models.** Week 0 is May 2013. Only top 5 clusters shown. Dotted lines indicate the beginning of the 2014 and 2015 outbreaks.

fit to the three main clusters for each human model and truncated each cluster to the year in which it caused its major outbreak (2015 for the primary cluster and 2014 for the secondary and tertiary clusters, dotted black lines Fig 3).

A fine scale dengue microsimulation model was then fit to each of the nine cluster datasets in Fig 3 using ABC SMC. All models offered improved fit (26.7–36.8% reduction in $d$) with each SMC round giving positive but diminishing improvements (S4 Fig). The radiation human movement model gave the best fit to the first cluster ($d$ = 2.66, Table 2) while the exponential model gave the best fit to clusters 2 and 3 ($d$ = 2.75 and $d$ = 2.95 respectively, Table 2).

**Table 2. Final values of the objective function (d) across and within clusters and its improvement between rounds 1 and 3 of sequential Monte Carlo rounds.**

| | Exponential movement | | Gravity movement | | Radiation movement | |
|---|---|---|---|---|---|---|
| | Final deviation | % improvement | Final deviation | % improvement | Final deviation | % improvement |
| Cluster 1 | 2.90 | 16.64 | 2.69 | 24.04 | 2.66 | 30.11 |
| Cluster 2 | 2.75 | 26.66 | 2.74 | 22.56 | 2.79 | 17.30 |
| Cluster 3 | 2.95 | 36.76 | 3.00 | 34.98 | 3.67 | 11.40 |

The posterior distribution of model parameters suggests differing patterns of transmission between cluster 1 (the largest cluster) and clusters 2 and 3 (Fig 4). The fitted model suggests that cluster 1 occurred among a group of less mobile individuals (mean 63% at-risk time spent at home) living in lower transmission intensity areas (mean 0.27 bites per day) but with a higher probability of reporting (0.15%).

In contrast, parameter values from cluster 2 and 3 fits suggest transmission among a more mobile set of the population (15% and 9% time at risk spent at home) in higher transmission intensity areas (0.70 and 0.68 bites per day) and a lower probability of reporting (0.04% and 0.03%). No clear correlation between transmission intensity in the current outbreak and historical transmission intensity was observed for any clusters (Transmission coefficient (correlation), Fig 4).

To assess the fitted model's ability to reconstruct the spatio-temporal patterns of transmission we compared the distribution of distances between cases in successive generations of

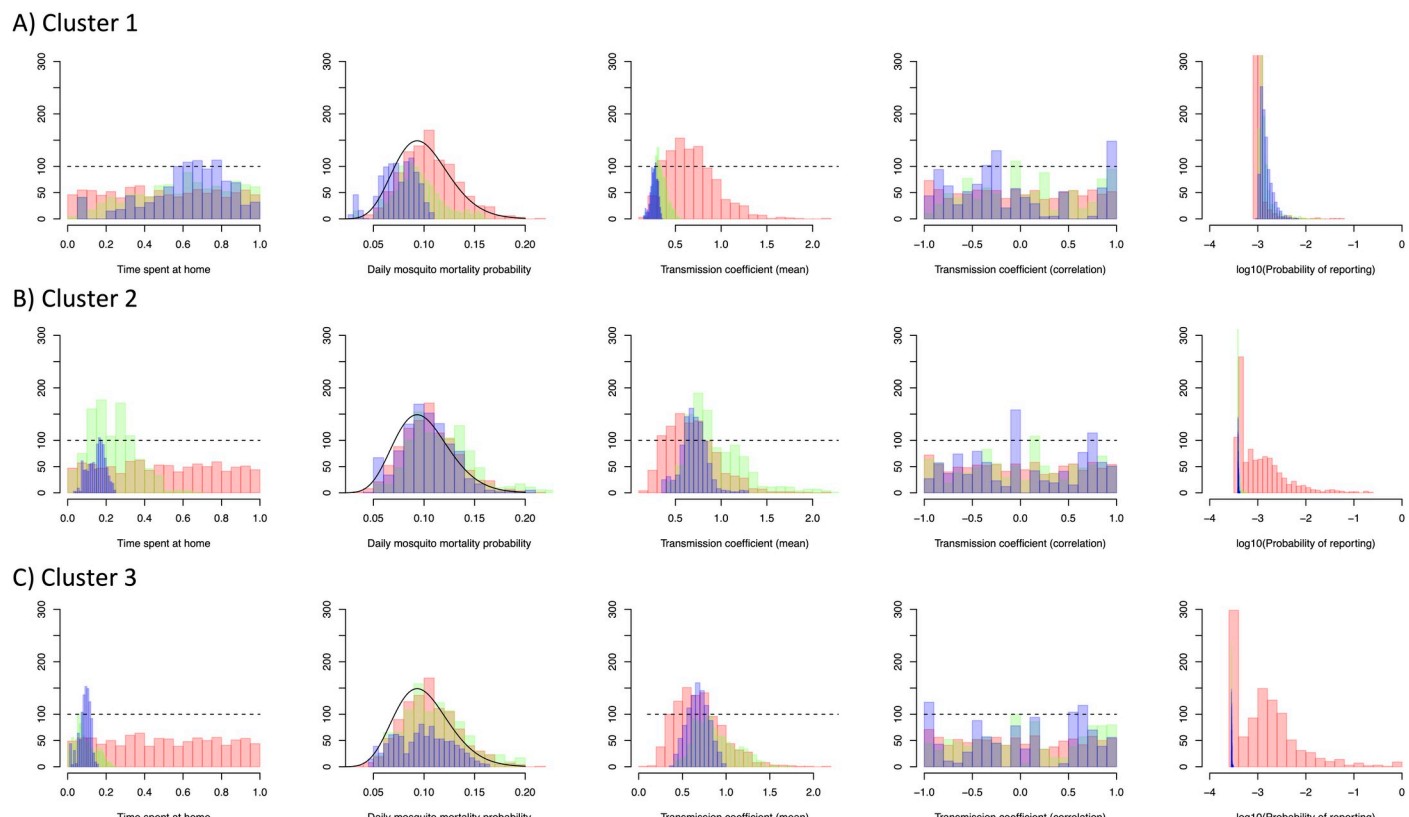

**Fig 4.** Histograms of the distribution of each model parameter using the best fitting human movement model (exponential for cluster 1, gravity for clusters 2 and 3) over successive sequential Monte Carlo rounds (Red = round 1, Green = round 2, Blue = Round 3). Probability of reporting is on a logarithmic scale (base 10).

transmission, i.e., dispersal kernels (Fig 5). Distance between cases two weeks apart were compared to approximate the serial interval of DENV. This analysis showed that the majority of cases in the next generation of transmission are likely to occur locally (84%, 80% and 86% for clusters 1–3) with long-distance dispersals orders of magnitude lower. The fitted models replicated comparable dispersal kernels, particularly for clusters 1 and 2. The models estimate comparable within patch transmission (77%, 70% and 61% for clusters 1–3) with a slight tendency to overestimate long distance transmission events (9000m+).

## CATI strategies can be effective in both the short and long term, but effectiveness is likely to be variable, especially with vector control

To assess the effectiveness of different CATI strategies, our model was run using an ensemble of posterior parameter values from each cluster to represent the likely effectiveness under real world conditions in Singapore where multiple chains of transmission often co-circulate. The model suggests that individuals living in areas that receive at least one CATI prophylactic drug dose at any point in the outbreak show strong protection against future infections over the duration of the drug's effective period. This level of protection, on average, exceeds the effective coverage level of the drug when deployed (median 100% predicted efficacy vs 81% drug effective coverage, Fig 6A), suggesting the non-treated individuals within the area receive some indirect protection from local, rapid control. Long-term effectiveness in treated patches declines but still maintains efficacy in most areas (median 50%). Effectiveness was, however, highly variable (95CIs span 0–1) and some patches see little or no impact. Multivariate regression analysis showed that patches in low population density areas had higher effectiveness than more densely populated areas and that more connected areas treated later in the outbreak had marginally higher effectiveness (Fig 6B).

For equivalent effective coverage and duration of efficacy, CATI with vector control is predicted to have lower, less persistent and considerably more variable effectiveness (effectiveness 63% short term, 0% long term, Fig 6A). Both short and long-term effectiveness of patches treated using a CATI strategy are likely to be lower than if vector control was consistently applied in these patches throughout the outbreak, although a CATI strategy would also use fewer resources. This occurs due to reinvasion of transmission into each patch, asymptomatic infection and, to a lesser extent, the additional delay in response time of vector control interventions (Fig 2). Applying both prophylactic drugs and vector control in a CATI strategy has the potential to marginally improve long-term effectiveness (53% vs 50%).

## CATI strategies with a large case radius can make up for late or low effectiveness responses

Next we tested the sensitivity of effectiveness of CATI strategies to I) the response radius around an index case, ii) the delay between case detection and intervention application and iii) the effectiveness of the intervention used. Due to the slow average spatial diffusion rate of the dengue outbreak, we observe only moderate sensitivity to delays in CATI response with only a 5–15% drop in effectiveness between a response that occurs on the day of detection compared to a response 2 weeks later (Fig 7A and 7B). Contrastingly, increasing the radius around the index case that interventions are deployed significantly increases overall effectiveness of the CATI strategy. We predict that a CATI response with a 100m radius delivered 2 weeks after detection will be more effective than a response with a 50m radius delivered on the day the case is detected, emphasising the importance of scale over speed in CATI response. CATI using vector control showed lower overall effectiveness and less sensitivity to implementation

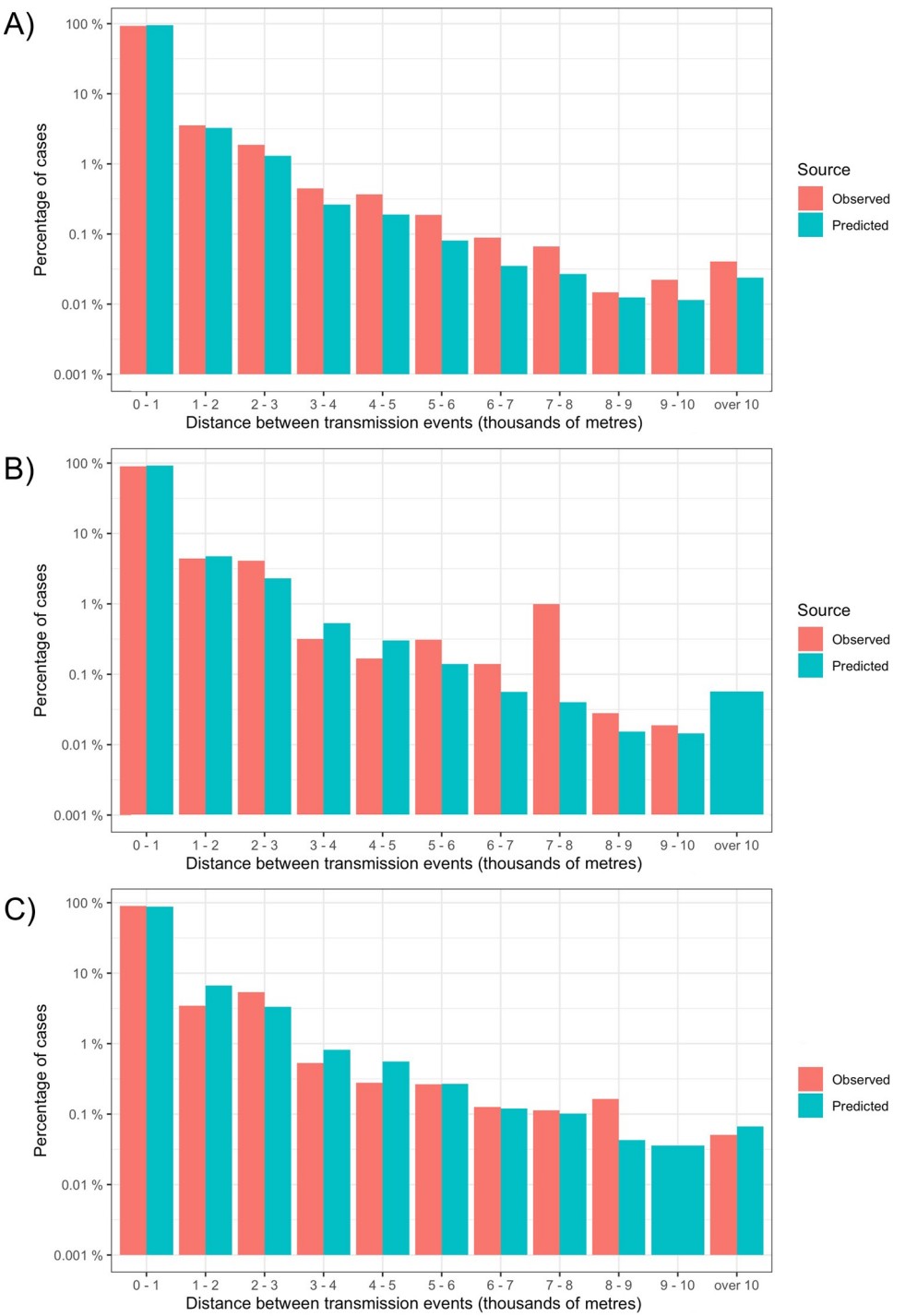

**Fig 5. Comparison of the spatial transmission kernels of observed cluster data and model predictions.** These histograms show the observed and model predicted distances between detected cases in the current week and cases two weeks (approximately one serial interval) ahead. Case distances are calculated as Euclidian distance to nearest neighbour with case location assigned to the midpoint of a 1km x 1km grid. Model predictions show the median of 10 runs from the best fitting human movement model (gravity for A and B, exponential for C). Number of cases shown on a log10 scale for cluster 1 (A), 2 (B) and 3 (C).

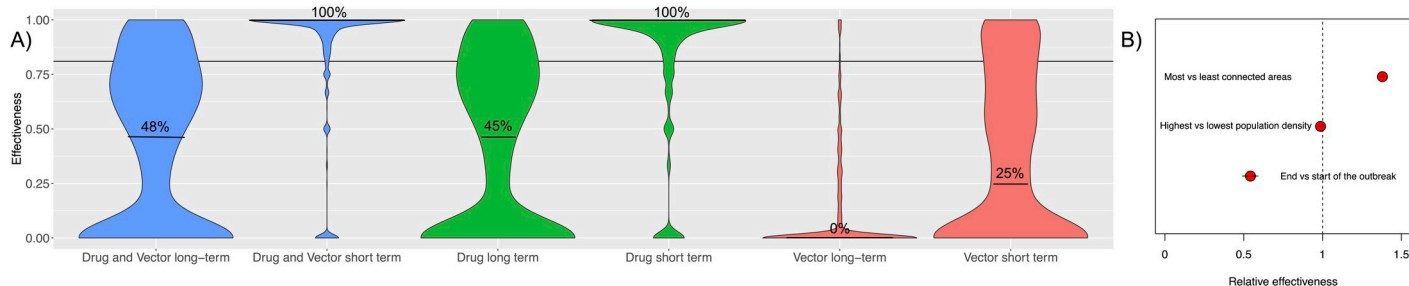

**Fig 6. Within patch effectiveness of case-area targeted interventions.** A) Effectiveness compares proportion of symptomatic cases in areas treated with prophylactic drugs and adult vector control over the short (30 day) and long (365 day) time periods. Both drugs and vector control are predicted to have 81% effective coverage (solid horizontal line) deployed the day after index case detection within a 1km radius of the index case, maintaining efficacy for 30 days. B) Relative effectiveness between the highest and lowest value pixels with respect to three characteristics (as predicted by a multivariate model fit to the long-term drug effectiveness results).

delay, but also showed substantial increases in effectiveness with larger intervention radii (Fig 7B).

Using an intervention with lower efficacy or lower coverage (i.e., lower effective coverage) predictably has a proportionally lower overall efficacy (Fig 7C and 7D). However, we predict that medium to high final effectiveness can still be achieved with low efficacy interventions when used in a CATI strategy with larger case radii (250-1000m, Fig 7C and 7D). This is because larger rings ensure hotspots of transmission are treated regularly and repeatedly, increasing the chance of effective control. This prediction does, however, assume that interventions fail at random as opposed to more fundamental barriers to effectiveness, e.g., insecticide resistance or hard-to-reach human or mosquito populations.

## CATI strategies are likely to be less effective in higher endemicity settings

Finally, we aimed to assess the generalisability of our findings to other dengue endemic settings outside Singapore where baseline transmission intensity is typically higher and case detection rates are lower (Fig 8). We predict that even modest increases in baseline transmission intensity ($\beta_\mu$) considerably reduce the ability of even the largest ringed CATI strategies (1000m) to control dengue outbreaks (doubling $\beta_\mu$ leads to ~ 2 times the number of infections). Reductions in case detection probability also significantly restrict the utility of CATI strategies for outbreak response. Reducing detection probability by 50% (relative to Singapore levels) leads to 10–69% more cases with the biggest increases in low endemicity settings such as Singapore.

## Discussion

CATI strategies are becoming increasingly popular for dengue control due to: i) a growing emphasis on outbreak prevention as the key goal of dengue control [41], ii) the failure of traditional control approaches to contain transmission long-term [24], iii) resource limitations and iv) increasing availability of data and data analytics platforms to predict and enable rapid precision responses [42]. There is, therefore, a growing need for a quantitative understanding of the strengths and limitations of CATI strategies to enable a more nuanced discussion of what role it can play in dengue control.

In this analysis we use a mathematical model, fit to a Singapore setting, to explore the impact of different CATI strategies. We find that controlling individual chains of transmission, e.g., within a household or local community, is likely to be highly time sensitive and prophylactic drugs could be a useful tool for extending this time window, particularly if people present

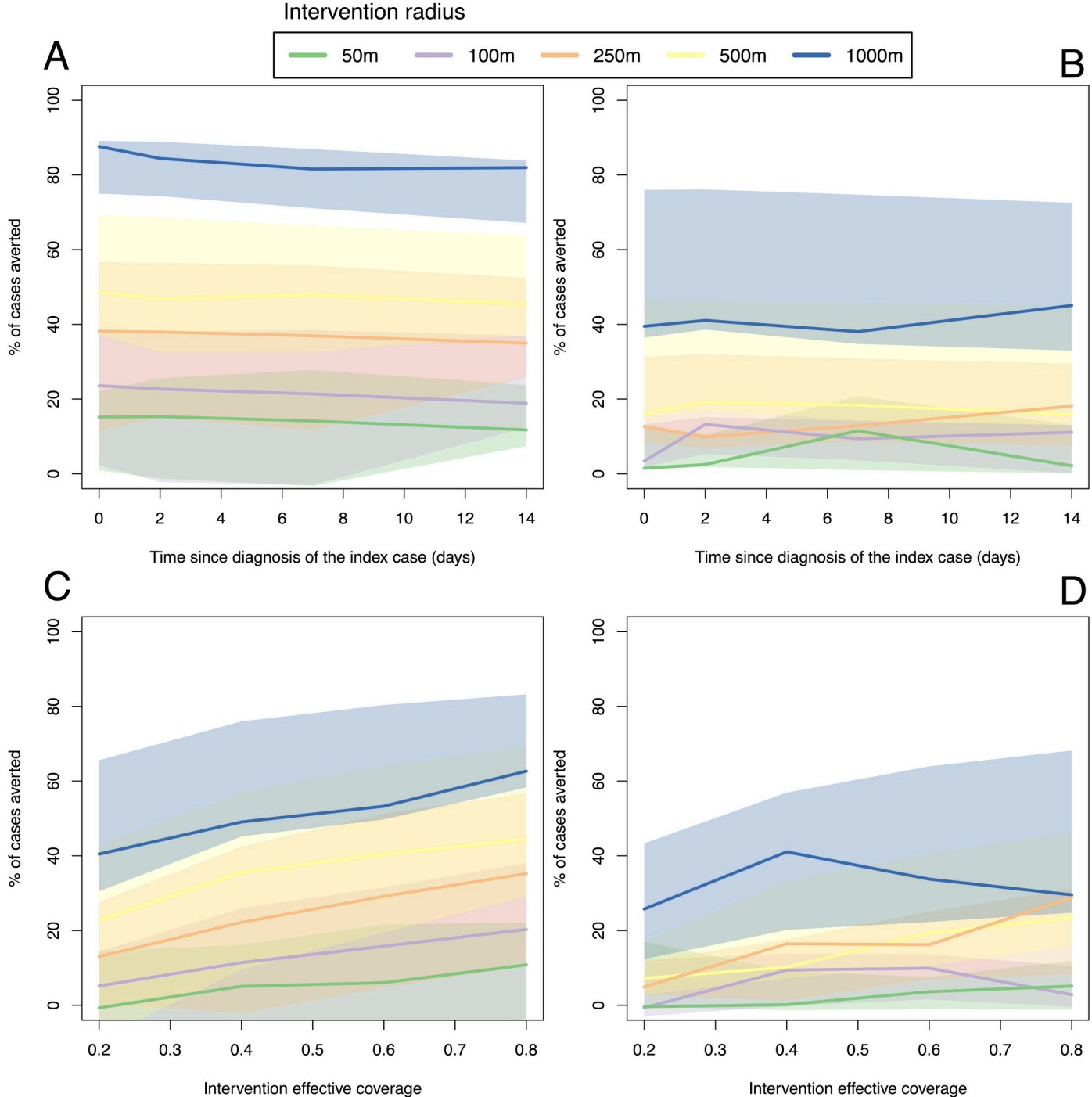

**Fig 7.** Scale, speed (A and B) and intervention effective coverage (C and D) required for a case-area targeted strategies with drugs (A and C) and vector control (B and D).

several days after the onset of symptoms. Following Phase 1 and 2 trials, more detailed modelling studies will be necessary to explore how the effectiveness of CATI strategies with prophylactic drugs changes with the specific characteristics (e.g., duration of efficacy) of each drug. Our results also have wider implications for index-case based intervention trial designs [43],

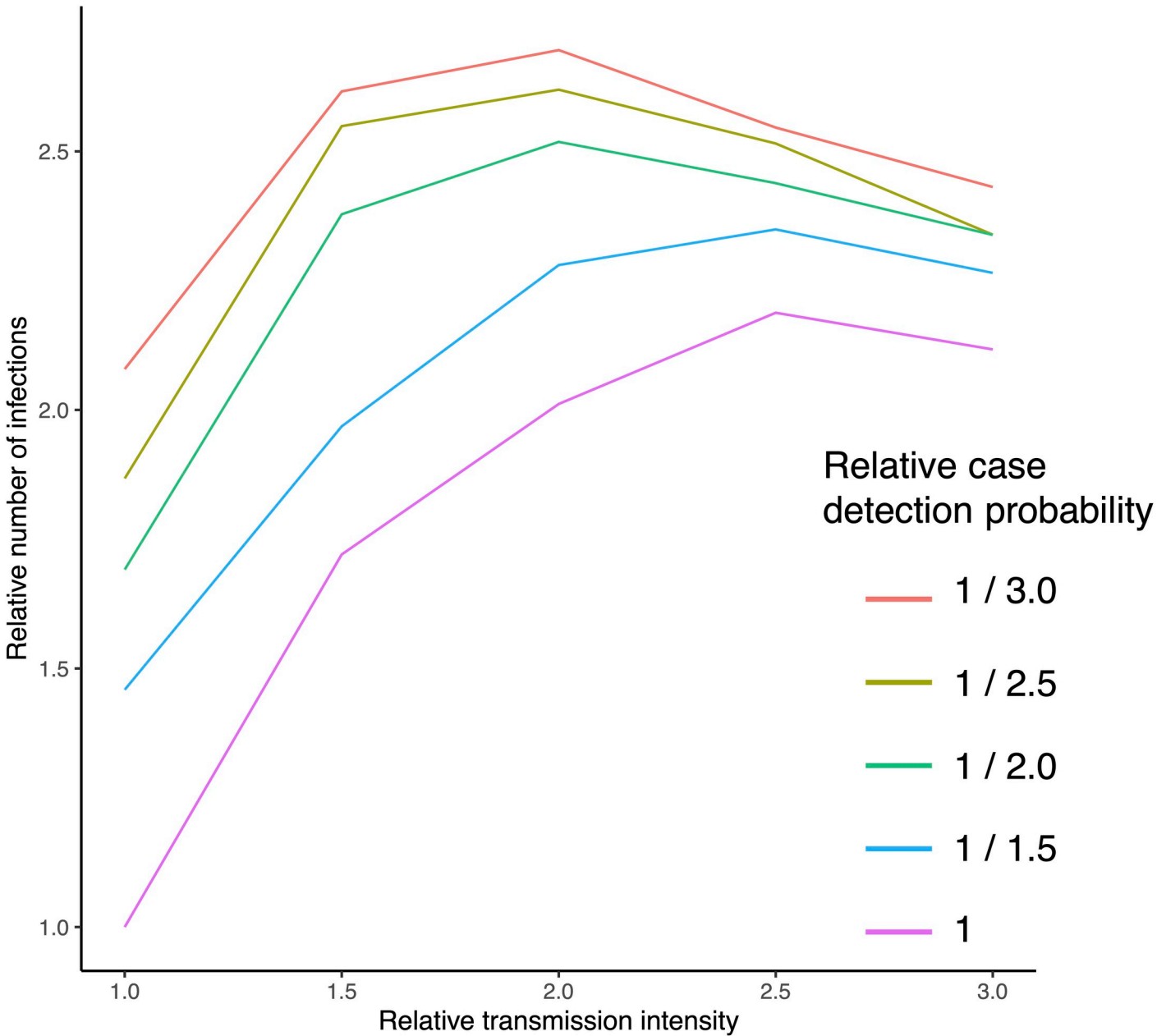

**Fig 8. Effectiveness of CATI strategies in higher endemicity settings.** All values proportional to Singapore parameters ($\beta_\mu$ and $\delta$). Assumes drug CATI strategy with 1km radius delivered the day of detection with 81% effective coverage lasting 30 days per response over a 1-year time horizon.

emphasising the need for rapid response to maintain intervention effect size, particularly for vector control tools that target adult mosquitoes [44]. We found that most areas treated one or more times using a CATI approach do see protection from dengue that lasts for the duration of the outbreak, especially in low density areas. Effectiveness of CATI strategies were found to be highly sensitive to the radius around the index case with radii 250m or larger giving the largest and most durable protection even if the efficacy of the interventions used is low. Finally, we estimate that CATI strategies will be highly sensitive to baseline transmission intensity and are only likely to be effective in areas that have already reduced transmission to lower levels

and have relatively high levels of case detection. This leads us to conclude that CATI strategies for dengue are likely to be effective, but are unlikely to be a precision, low resource requirement solution that is universally applicable. The need to have already achieved low levels of baseline transmission intensity, have high rates of case detection and be able to repeatedly deploy interventions at broad spatial scales suggest that CATI strategies will be best suited to non-endemic areas or as a "final-mile" approach in the latter stages of a wider dengue elimination programme. Similar conclusions have been reached on the role of reactive case detection and focal mass drug administration in malaria elimination [45, 46]—a vector-borne disease that is complicated by similar issues of asymptomatic infection and mobility of infected individuals. This also highlights the need for targeting preventative interventions such as environmental clean-up campaigns and vaccines (e.g., CDY-TDV) in high density, high transmission areas which may then allow CATI to contain outbreaks where it previously could not.

The predicted between-cluster variability in effectiveness of CATI also suggests that such strategies may need to be flexible and adaptable, changing index case radius, timeliness and even mixtures of interventions in response to the unfolding dengue epidemic. CATI strategies would, therefore, be best implemented alongside additional serological human or entomological sampling to further understand where and why effectiveness changes.

Our findings are subject to a number of limitations of our chosen model structure, fitting procedure and generalisability. Our model likely underrepresents geographic heterogeneity in mosquito-human contact, particularly in Singapore where small-scale construction sites are known to be significant contributors to overall transmission [47]. Higher geographic heterogeneity would make individual hotspots harder to control, but potentially more susceptible to focal targeting if they can be successfully identified. Over the time period of our data entomological surveys were only available from a small number of sentinel sites that were sufficient to measure variation in mosquito abundance over time, but not space at a sufficiently high resolution. If entomological data were available at higher spatial resolution the model may have been able to better capture the observed spatial heterogeneity in dengue cases [48]. Our model also demonstrated variable ability to fit different case clusters in Singapore, particularly outbreaks with more geographically restricted ranges. More work is needed to understand if and why the spatial spread of some epidemics differs. This may require developing a multi-serotype model to reflect co-circulation of multiple DENV serotypes and the complex pattern of human immunity that they impose in settings like Singapore. Integrating human movement data from, for example call data records [49], instead of relying on human moment models may also help better characterise movement heterogeneity across Singapore. Developing multiple-infection models with realistic population dynamics will also be important for projecting the longer-term reductions in disease burden attributable to CATI. This is important as it remains unknown whether the infections CATI averts are truly averted or just delayed until later in an individual's lifetime.

We also did not consider spatio-temporal variability in vector control when fitting our model. Our model placed no restrictions on the frequency within which interventions could be re-applied. Continual dosing of dengue prophylactic drugs over multiple months may not be feasible or practical. The model also assumed random efficacy and coverage of the intervention meaning multiple rounds of drug dosing or vector control boosted effective coverage and led us to the conclusion that large radii CATI strategies would be particularly effective. In reality there may be a cap on intervention effective coverage due to hard-to-reach populations/vector habitats or individuals/individual houses ineligible for drug dosing.

CATI strategies are becoming increasingly prevalent components of wider dengue control strategies. Here we aimed to quantitatively assess their potential effectiveness and assess their sensitivity to a range of operational and epidemiological parameters. This emphasised the

importance of using broad radii around index cases and the need to continue other interventions to keep baseline transmission intensity low and case detection rates high. If this can be achieved, CATI strategies with drugs or vector control could play an important role dengue control efforts.

## Supporting information

**S1 Fig. Contains a plot of the entomological and epidemiological data over time.**
(DOCX)

**S2 Fig. Fit of the model to the data on human infectiousness over time.**
(DOCX)

**S3 Fig. Map of estimated seroprevalence at baseline in Singapore.**
(DOCX)

**S4 Fig. Model fit to each case cluster over successive rounds of Sequential Monte Carlo sampling.**
(DOCX)

**S1 Text. Describes how the population base maps were generated.**
(DOCX)

**S2 Text. Equations for human movement models.**
(DOCX)

**S1 Table. Sensitivity of clustering algorithm to choice of linkage break threshold.**
(DOCX)

## Author Contributions

**Conceptualization:** Oliver J. Brady, Marnix Van Loock, Guillermo Herrera-Taracena, Joris Menten, W. John Edmunds, Stéphane Hué, Martin L. Hibberd.

**Data curation:** Yalda Jafari, Shuzhen Sim, Lee-Ching Ng.

**Formal analysis:** Oliver J. Brady, Yalda Jafari.

**Funding acquisition:** Stéphane Hué, Martin L. Hibberd.

**Investigation:** Oliver J. Brady, Marnix Van Loock, Guillermo Herrera-Taracena, Joris Menten, W. John Edmunds, Shuzhen Sim, Lee-Ching Ng, Stéphane Hué, Martin L. Hibberd.

**Methodology:** Oliver J. Brady, Adam J. Kucharski, Sebastian Funk, Yalda Jafari, W. John Edmunds, Shuzhen Sim, Lee-Ching Ng, Stéphane Hué, Martin L. Hibberd.

**Supervision:** Stéphane Hué, Martin L. Hibberd.

**Validation:** Oliver J. Brady, Adam J. Kucharski, Sebastian Funk, Yalda Jafari, Shuzhen Sim, Lee-Ching Ng.

**Visualization:** Oliver J. Brady.

**Writing – original draft:** Oliver J. Brady, Stéphane Hué, Martin L. Hibberd.

**Writing – review & editing:** Oliver J. Brady, Adam J. Kucharski, Sebastian Funk, Yalda Jafari, Marnix Van Loock, Guillermo Herrera-Taracena, Joris Menten, W. John Edmunds, Shuzhen Sim, Lee-Ching Ng, Stéphane Hué, Martin L. Hibberd.

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
