## [Decision Letter · Decision Letter 0]

24 Feb 2021

Dear Dr Brady,

Thank you very much for submitting your manuscript "Case-area targeted interventions (CATI) for reactive dengue control: modelling effectiveness of vector control and prophylactic drugs in Singapore" for consideration at PLOS Neglected Tropical Diseases. As with all papers reviewed by the journal, your manuscript was reviewed by members of the editorial board and by several independent reviewers. In light of the reviews (below this email), we would like to invite the resubmission of a significantly-revised version that takes into account the reviewers' comments. 

We cannot make any decision about publication until we have seen the revised manuscript and your response to the reviewers' comments. Your revised manuscript is also likely to be sent to reviewers for further evaluation.

Sincerely,

Hannah E Clapham

Associate Editor

Nigel Beebe

Deputy Editor

Reviewer's Responses to Questions

**Key Review Criteria Required for Acceptance?**

**Methods**

-Are the objectives of the study clearly articulated with a clear testable hypothesis stated?

-Is the study design appropriate to address the stated objectives?

-Is the population clearly described and appropriate for the hypothesis being tested?

-Is the sample size sufficient to ensure adequate power to address the hypothesis being tested?

-Were correct statistical analysis used to support conclusions?

-Are there concerns about ethical or regulatory requirements being met?

Reviewer #1: I have several questions regarding the validity and applicability of the analysis.

1) GAI index used for mosquito abundance, how is this sampled? Can it affect results if unequal sampling has been carried out across Singapore?

2) Patches, is 1km x 1km appropriate? A lot of area based heterogeneity, which may not be represented within patch dynamics if grids are just defined by "squares". Furthermore, many barriers exist within Singapore, some porous, others not so porous. These will significantly affect spatial mosquito distributions where 1km will not capture small scale sudden outbreaks - by the time an area of 1km is notifying a case surge, it is a fire fighting regional effort to control mosquito populations and transmission and therefore outbreaks are not really averted. Microscale treatment around houses would not avert an outbreak.

3) Calibration of model to all years? Or only within epidemic season? How was model fit overall on epidemic trajectory? Does it replicate observed seasons over time? Measure of correlation between fit and observation (R^2) possible? 

4) How will between serotype interactions affect the results? Singapore is hyper-endemic. Serotype switching is well documented to pre-empt looming epidemics.

5) The differences in movement types (gravity, radiation, exponential) has been well studied elsewhere. Of great usefulness would be to use a fitted movement distribution for mosquitoes rather than theoretical ones, in a smaller scale study area/areas where mosquito gravitrap data has been collected. This would be much more convincing of CATI's usefulness.

6) In reading CATI: https://journals.plos.org/plosmedicine/article?id=10.1371/journal.pmed.1002509, Singapore has referenced "CATI" simply as vector control and has been carrying this out since vector control practices began. The use of a prophylatic drug (which has been shown to be applicable to cholera) has a myriad of issues for dengue in terms of lifespan of the drug and whether such a drug will be effective in such a scheme in terms of deployment - if a prophylatic is shown to be successful, population wide implementation or the arrival of a new serotype would be much more advantageous as a signal of an upcoming epidemic.

Reviewer #2: The model design makes it clear how CATI effectiveness will be measured, and all data is made readily available. There are some gaps not addressed when comparing the model structure to the DENV transmission pattern that are explained in the notes attached. In essence, not allowing individuals to become susceptible again may obscure the relationship between DENV prevalence and dengue burden. Given the significant baseline seroprevalence (46%), individuals appear likely to be infected with DENV more than once regardless of number of circulating serotypes. 

Beyond this, when considering whether CATI strategies are effective, it may be nice to expand on how prevention of onward transmissions relates to actual reduction in disease burden. This is somewhat dependent on the ability to number infections, but rates of disease based on infection number are described here. https://journals.plos.org/plosmedicine/article?id=10.1371/journal.pmed.1002181

Reviewer #3: -Are the objectives of the study clearly articulated with a clear testable hypothesis stated? 

Yes

-Is the study design appropriate to address the stated objectives?

Yes

-Is the population clearly described and appropriate for the hypothesis being tested?

Yes

-Is the sample size sufficient to ensure adequate power to address the hypothesis being tested?

Yes

-Were correct statistical analysis used to support conclusions?

Yes

-Are there concerns about ethical or regulatory requirements being met?

No

**Results**

-Does the analysis presented match the analysis plan?

-Are the results clearly and completely presented?

-Are the figures (Tables, Images) of sufficient quality for clarity?

Reviewer #1: (No Response)

Reviewer #2: From the model fit on, results match what is described in the methods and background section. The section starting on line 403 is confusing as is. It seems that 100% effectivity is assumed for both vector and prophylactic solutions, an unrealistic standard in dengue prevention. If this is instead intended to describe model behavior or assumptions, the heading on line 400 should be changed to reflect this. 

Figures are easy to understand and work well in context. Figure 8’s y-axis should be changed to say “Relative number of infections”

Reviewer #3: -Does the analysis presented match the analysis plan?

Yes

-Are the results clearly and completely presented?

Generally Yes

-Are the figures (Tables, Images) of sufficient quality for clarity?

I have included some questions on some of the figures.

**Conclusions**

-Are the conclusions supported by the data presented?

-Are the limitations of analysis clearly described?

-Do the authors discuss how these data can be helpful to advance our understanding of the topic under study?

-Is public health relevance addressed?

Reviewer #1: (No Response)

Reviewer #2: Results are contextualized well, and limitations of model are appropriately considered. I would suggest that locations that are fit for CATI be explicitly mentioned, and put in contrast with locations where CATI is not appropriate. I noted this in methods, but I believe that rates of symptomatic cases by serial infection and the CYD-TDV vaccine should be mentioned here as well.

Reviewer #3: -Are the conclusions supported by the data presented?

Generally yes. I have made some suggestions to improve the strength of the conclusions.

-Are the limitations of analysis clearly described?

Yes

-Do the authors discuss how these data can be helpful to advance our understanding of the topic under study?

Yes

-Is public health relevance addressed?

Yes

**Editorial and Data Presentation Modifications?**

Reviewer #1: (No Response)

Reviewer #2: Accept data presentation as presented.

Reviewer #3: Minor revision

**Summary and General Comments**

Reviewer #1: (No Response)

Reviewer #2: This paper does a good job at describing CATI and its appropriate role in dengue control. Its model supports the conclusions drawn and its limitation and sensitivities are described well.

Reviewer #3: (No Response)

PLOS authors have the option to publish the peer review history of their article (what does this mean?). If published, this will include your full peer review and any attached files.

Reviewer #1: No

Reviewer #2: Yes: Magdalene Walters

Reviewer #3: No
---

## [Editor Report · Decision Letter 1]

14 Jun 2021

Dear Dr Brady,

We are pleased to inform you that your manuscript 'Case-area targeted interventions (CATI) for reactive dengue control: modelling effectiveness of vector control and prophylactic drugs in Singapore' has been provisionally accepted for publication in PLOS Neglected Tropical Diseases.

Best regards,

Hannah E Clapham

Associate Editor

Nigel Beebe

Deputy Editor

---

## [Editor Report · Acceptance letter]

9 Jul 2021

Dear Dr Brady,

We are delighted to inform you that your manuscript, "Case-area targeted interventions (CATI) for reactive dengue control: modelling effectiveness of vector control and prophylactic drugs in Singapore," has been formally accepted for publication in PLOS Neglected Tropical Diseases.

Best regards,

Shaden Kamhawi

co-Editor-in-Chief

Paul Brindley

co-Editor-in-Chief
